# Biosynthesized Gold, Silver, Palladium, Platinum, Copper, and Other Transition Metal Nanoparticles

**DOI:** 10.3390/pharmaceutics14112286

**Published:** 2022-10-25

**Authors:** Piotr Roszczenko, Olga Klaudia Szewczyk, Robert Czarnomysy, Krzysztof Bielawski, Anna Bielawska

**Affiliations:** 1Department of Biotechnology, Medical University of Bialystok, Kilinskiego 1, 15-089 Bialystok, Poland; 2Department of Synthesis and Technology of Drugs, Medical University of Bialystok, Kilinskiego 1, 15-089 Bialystok, Poland

**Keywords:** metal nanoparticles, green synthesis, anticancer activity, antibacterial activity

## Abstract

Nanomedicine is a potential provider of novel therapeutic and diagnostic routes of treatment. Considering the development of multidrug resistance in pathogenic bacteria and the commonness of cancer, novel approaches are being sought for the safe and efficient synthesis of new nanoparticles, which have multifaceted applications in medicine. Unfortunately, the chemical synthesis of nanoparticles raises justified environmental concerns. A significant problem in their widespread use is also the toxicity of compounds that maintain nanoparticle stability, which significantly limits their clinical use. An opportunity for their more extensive application is the utilization of plants, fungi, and bacteria for nanoparticle biosynthesis. Extracts from natural sources can reduce metal ions in nanoparticles and stabilize them with non-toxic extract components.

## 1. Introduction

Since ancient times, gold colloidal solutions have been considered to have healing properties against heart diseases, cancer, or epilepsy. The development of nanotechnology as science was initiated by Michael Faraday in 1857 when he gave a lecture on the experimental relationship between gold and light. This was the first report on the interaction of light and gold, and other metals at the nanometer size. The current shape of nanotechnology began with the lecture “There’s plenty of room at the bottom” given by Richard Feynman in 1959 [1]. The first use of nanoparticles (NPs) in biomedical applications was proposed by Paul Ehrlich as a way to improve drug therapy as a carrier for therapeutic substances. The systems he developed were called magic bullets [2].

There have been attempts to create such particles using bacterial, plant, or fungal extracts (Figure 1) [3]. Moreover, there are efforts to synthesize nanoparticles using living organisms. Earthworms can generate cadmium telluride (CdTe) quantum dots, which may be applied in live-cell imaging. Several formulations are based on heavy metals, with their safety verified by the detection of iron oxide NPs in the brain. Platinum, for instance, is used to treat many types of cancer [4] or gadolinium is used in contrast agents [5]. The physiological environment establishes the conditions for the transition of these formulations into metal nanoparticles. Biotransformation of dietary zinc ions to ZnO NPs can occur spontaneously in animal models. Remarkably, NPs based on noble metals—such as Au, Pt, and Ag—can spontaneously absorb glutathione (GSH) through the sulfhydryl group, which is relevant to the role of reduced GSH in the mechanism of multidrug resistance of neoplasms [6].

The size of nanoparticles is crucial to their functionality. This is related to the particle surface area, which is inversely proportional to their size. The smaller the particle, the greater its surface area and the greater its ability to bind to a biological structure. NPs are used both in imaging and in the transfer of an active particle to the affected tissue. Nanocarriers can be polymer conjugates, liposomes, dendrimers, peptides, carbon, silica, metal, or metalloid nanoparticles. They can improve treatments, resulting in redrafted healthcare delivery and offset the side effects of standard therapy [7].

Transition metal nanoparticles induce oxidative stress due to their small size and the reactive surface area leading to cellular reactions. NPs enter cells by endocytosis and mainly accumulate in lysosomes and mitochondria, impairing their physiological functions. The acidic environment in lysosomes promotes the dissolution of nanoparticles to ions, which participate in Fenton or Haber-Weiss-type reactions. This subsequently leads to the generation of reactive oxygen species in cells, which is most likely the main mechanism of the cellular response [8].

Nanoparticles can elevate the physicochemical parameters of any widely used drug. Nanotechnology overcomes the limitations of conventional substances, such as insufficient membrane trafficking or particle distribution. They improve the stability and solubility of compounds and increase the drug’s residence time in circulation [9]. An additional benefit is delivering the medication in a targeted manner to a specific, disease-affected tissue or organ. The extended release of the therapeutic substance contributes to a decrease in the drug dose frequency for the patient [10].

Cancer is a core cause of death worldwide, accounting for nearly 10 million deaths in 2020 [11]. Thus far, there is no effective and safe cure for the treatment of cancer. One future trend in treating cancerous lesions is nanoparticles (NPs), which exhibit anticancer activity. Their use in the treatment of oncological diseases allows effective tumor reduction with decreased toxicity to healthy tissues.

The increase in morbidity and mortality resulting from bacterial infections is caused by the emergence of drug-resistant microorganisms and poses a real public emergency. The application of nanoparticles as new biomaterials is currently gaining attention. Nanoparticles are a promising therapeutic option for the treatment of drug-resistant infections. Nanoparticles of metals and their oxides appear to be the most effective [12].

Antibiotics with anticancer effects, such as doxorubicin, bleomycin, or mitomycin C, are known and widely used in the medical field [13]. Bacterial anticancer agents such as antibiotics, polyketides, bacteriocins, or non-ribosomal peptides act by inducing apoptosis, and necrosis, reducing angiogenesis or blocking essential signaling pathways [14].

Biomedical applications might also include the use of green nanoparticles as detectors for substances in biological samples. This is essential, as there are efforts to improve our understanding of the molecular mechanisms underlying diseases [15]. More and more sensitive detectors may lead to a broader understanding and earlier diagnosis of diseases. 

Green nanoparticles are a more environmentally safe option for treating cancer and antibiotic-resistant bacteria or creating sensors to detect substances in biological samples. As an indirect biomedical application, nanoparticles synthesized with plants, bacteria [16], waste [17], or fungi can also be used as catalysts in the synthesis of new substances with anticancer and antimicrobial properties. By using these catalysts, we are able to reduce environmental pollution—a challenge in our times.

The biomedical use of green nanoparticles is associated with a risk of side effects after exposure. Nanoparticles have the potential to accumulate, and there is a possible risk of systemic toxicity [18]. The complete mechanism of action and toxicity of NPs-based formulations should be evaluated before introducing nanoparticles in clinical trials.

## 2. Nanoparticle Synthesis and Analysis

Natural extracts contain numerous organic compounds, such as alkaloids, flavonoids, proteins, or polysaccharides, which are utilized in the synthesis of nanoparticles. They lead to the reduction of metal ions in NPs and are stabilizing agents. The biosynthesis mechanism involves reducing metal ions to a zero oxidation state. Owing to the various phytochemicals in the plant extract, functional groups of amides and amino carboxyl can be responsible for the stabilization of synthesized nanoparticles (Figure 2) [19].

Nanoparticle characterization has a significant role in the development of nanotechnology. Research focuses on two important points, the determination of particle size and surface charge of nanoparticles, in nanoparticle quality control. Numerous important biological properties of nanoparticles depend on size, and these properties are not effective until particles are reduced to nanometer sizes. Therefore, determining the particle size of nanoparticles is almost the most important characterization parameter An important aspect affecting the effectiveness of nanoparticles is also their morphology. Depending on the shape of the nanoparticles and the development of the surface area, the distribution parameters in the bloodstream, as well as the reactivity, vary [20]. There are many methods by which we can characterize the properties of nanoparticles. Fourier-transform infrared (FT-IR) spectroscopy demonstrates ligand binding, surface composition, and drug structure in nanoparticles. Zeta potential and Dynamic light scattering (DLS) show nanoparticle surface change and agglomeration state. UV-Vis discloses optical properties and concentration, and SEM, TEM, and NMR show nanoparticle size and shape [21]. The different methods for the analysis of biosynthesized nanoparticles are summarized in Table 1.

## 3. Gold Nanoparticles

### 3.1. Anticancer Activity

#### 3.1.1. Plants

The biosynthesis of gold nanoparticles in the presence of perlite using *Allium Fistulosum* L. extract as a stabilizing and reducing agent resulted in Au/perlite nanoparticles with potential applications in PTT therapy. ROS generation by the synthesized nanoparticles was evaluated by DPPH assay. The results demonstrated that Au/perlite NPs had a radical scavenging capacity of nearly 42% within 30 min. The PTT efficacy of Au/perlite NPs was tested on the MCF-7 cell line by cell cycle and cell viability analysis. It is noteworthy that a dramatic decrease in survival occurred from 6 to 10 min. Cell cycle results showed an apoptotic cell count of 85% after 10 min of laser irradiation [22].

Gold nanoparticles were successfully synthesized from aqueous extracts of the traditional Chinese medicine *Dendrobium officinale* (DO) using green chemistry rules. To evaluate antitumor efficacy, in vitro studies were performed on HepG2 and L02 cell lines. Immunohistochemical analysis was also performed in vivo. Do-Au NPs had better antitumor efficacy compared with DO extract alone [24].

The synthesis of Au NPs proceeded with aqueous and ethanolic extract of *Taxus baccata*. An MTT assay was performed on cell lines such as Caov-4, MCF-7, and HeLa to evaluate the effect of nanoparticles. Furthermore, exposure of cells to Au NPs obtained from *T. baccata* confirmed that cell death in a caspase-independent manner is an antitumor mechanism with enhanced efficacy. This was confirmed using flow cytometry and real-time PCR [25].

Extracts from *Camellia sinensis*, *Syzygium aromaticum, Coriandrum sativum*, *Artabotrys hexapetalus*, *Mentha arvensis*, *Phyllanthus amarus*, and *Mimusops elengi* were used for nanoparticle biosynthesis. The synthesized Au NPs were found to exhibit antitumor activity against the human MCF-7 cell line. Notably, Au NPs at a minimum concentration of 2 µg/mL were as effective as chemotherapy drugs. Moreover, the effectiveness of the nanoparticles was directly proportional to their concentration [41].

*Viola betonicifolia* is a resource of various phytochemicals—such as alkaloids, flavonoids, tannins, and triterpenoids—which exhibit biological activity in multiple pharmacological routes [42]. VB-Au nanoparticles were synthesized using leaf extract, which possesses both reducing and stabilizing functions. The VB-Au NPs were characterized using spectroscopic techniques. The antimicrobial properties of VB-Au NPs and in vitro cytotoxic properties against MCF-7 cancer cells and human mesenchymal stem cells hMSCs were investigated. The VB-Au NPs exhibited excellent antimicrobial and biofilm inhibitory properties against the examined microbial species in comparison with the plant leaf extract and commercially purchased Au NPs. In addition, they also revealed significant antioxidant potential. The nanoparticles exhibited good compatibility with hMSCs and demonstrated promising cytotoxic potential against MCF-7 cancer cells in comparison with commercial Au NPs and the extract itself. These properties confirm the synergistic effect of the physical properties of the nanoparticles and the compounds from the extract adsorbed on their surface [43].

*Mentha piperita* leaf extract has properties that allow the reduction of chloroauric acid to gold nanoparticles [44]. The study proved that the synthesized Au NPs had a hexagonal structure with a diameter of approximately 78 nm. The biosynthesized Au NPs exhibited significant activity against the tumor cell lines MDA-MB-231 and A549 compared with the normal cell line 3T3-L1. The anti-inflammatory and analgesic effects were investigated in a Wistar rat model. Au NPs provided positive results for both lines of activity, although their potency was decreased compared with drugs currently applied in the pharmaceutical industry [45].

#### 3.1.2. Fungi

*Cordyceps militaris* is a species of fungus used in China as part of alternative medicine. Gold nanoparticles were synthesized with *C. militaris* extract. The stability and integrity of the Au NPs were confirmed by studies. The size of the gold nanoparticles was approximately 1520 nm; FT-IR results indicated that the gold nanoparticles incorporated hydroxyl groups and alkenes. The effect of gold nanoparticles from *C. militaris* on cells of the hepatocellular carcinoma line HepG2 was examined. Au NPs revealed a value IC_50_ between 10 µg and 12.5 µg/mL. The mechanism of action of the synthesized nanoparticles is rooted in the formation of ROS and alteration of the mitochondrial membrane potential. The initiation of apoptosis through the activation of Bax, Bid, and caspase was reported [23].

#### 3.1.3. Pure Substances

The application of toxic chemicals for stabilization—such as borohydride derivatives or sodium citrate—is a significant limitation of chemical gold nanoparticles in therapeutic applications such as photothermal therapy (PTT). Natural compounds like apigenin (API) can be a substitute in the biosynthesis of gold nanoparticles [46]. API@Au NPs with a size of 19.1 nm and a surface charge of 4.3 mV were synthesized using a straightforward and eco-friendly method. The stability of API@Au NPs was verified by UV-Vis spectroscopy and Raman and FTIR spectroscopy. Chemical binding of API on the surface of API@Au NPs via hydroxyl and carbonyl functional groups was confirmed to be the root cause of their stability compared with chemically synthesized and citrate-stabilized nanoparticles. The effect of the nanocomposite was investigated on mouse fibroblastic carcinoma (L929) and colon cancer (CT26) lines. Flow cytometry analysis indicated that the cell death mechanism was mainly apoptosis. Cell death induced by conventional gold nanoparticles in PTT induces necrosis in the majority of cases [47].

### 3.2. Antibacterial Activity

#### Bacteria

Gold nanoparticles were successfully synthesized using Antarctic bacteria. The process took place at different temperatures ranging from 4°C to 37 °C using a bacterial isolate obtained from an Antarctic lake. Phylogenetic and biochemical analysis revealed that the collected isolate contained *Bacillus* sp. Biosynthesis proceeded at each tested temperature but only in the logarithmic phase of bacterial growth. The biosynthesized gold nanoparticles exhibited antibacterial activity against sulfate-reducing bacteria (*Desulfovibrio* sp.). At a concentration of 200 µg/mL Au NPs, the growth rate was reduced by 12%, while the sulfate-reducing activity was reduced by 7%. The comet assay showed that a genotoxic effect was responsible for the inhibition of growth and sulfide production [32].

### 3.3. Multidirectional Activity

#### Waste Materials

Gold nanoparticles were biosynthesized using chitosan derived from squid shell waste. TEM analysis showed that the biosynthesized Au NPs had a spherical shape with a size between 80 and 82 nm. The Au NPs demonstrated activity against selected G+ and G- bacteria and revealed antifungal activity. In addition, the cytotoxic effect of the synthesized Au NPs was investigated against MCF-7 cell lines. The MTT assay revealed an IC50 of 250 μg mL^−1^. Furthermore, staining of MCF-7 cells with acridine orange and ethidium bromide showed that Au NPs lead to cell death by apoptosis [34].

### 3.4. Toxicity

#### Bacteria

An attempt was made to synthesize gold nanoparticles using the bacteria *Rastrelliger kanagurta*, *Panna microdon*, and *Selachimorpha* sp. Ultraviolet (UV)-visible spectroscopic analysis, Fourier transform infrared spectroscopy, X-ray diffraction, transmission electron microscopy, and scanning electron microscopy were used to characterize the gold nanoparticles. Based on TEM and SEM, the size of NPs was estimated to range from 45–80 nm. To evaluate the toxicity of the synthesized Au NPs, zebrafish was used as an animal model. The synthesized gold nanoparticles at a concentration of 100 μg/mL revealed high toxicity to zebrafish larvae. After 72 h, all experimental models exhibited mortality. No mortality was observed in the control groups. Gold nanoparticles demonstrated good antimicrobial and antitubercular activity against *Staphylococcus aureus*, *Micrococcus luteus*, *Streptococcus mutans*, *Pseudomonas fluorescens*, *Proteus* sp., and. *M. tuberculosis*. It was reported that gold nanoparticles have relatively low toxicity and, therefore, can be applied in pharmacy and medicine [48].

## 4. Silver Nanoparticles

### 4.1. Anticancer Activity

#### Plants

Extracts from *Artemisia peelscens* were used to produce *Artemisia*-Ag NPs. The nanoparticles were extensively characterized and screened on cancer cell lines—including HeLa and MCF-7. *Artemisia*-Ag NPs demonstrated inhibition of cancer cell growth and arrest in the G1 phase of the cell cycle. RNA sequencing was conducted as well, which revealed the potential of *Artemisia*-AgNPs in cancer research. Components of the plant extract improved the efficacy of Ag NPs [49].

Medicinal herbs have the potential to be used for the biosynthesis of silver nanoparticles and provide an alternative to traditional chemical or physical methods used at present [50]. *Matricaria recutita* was found to have unique properties, including anticancer activity. Ag NPs were biosynthesized using aqueous extracts from *M. recutita* against A549 lung cancer cells. UV-Vis spectra showed maximum absorption of bio-Ag NPs at 430 nm. The crystal structure was confirmed by XRD; EDX analysis showed the presence of Ag as a constituent element. FT-IR results also proved the synthesis of Ag NPs using the plant extract. Spherical shapes of Ag NPs were formed with an average diameter of 45.12 nm and a zeta potential value of 34 mV, which was confirmed by TEM and FE-SEM. MTT assay showed a dose- and time-dependent cytotoxic effect against A549 lung cancer cells. Furthermore, the apoptotic effect of AgNPs was demonstrated by DAPI staining and flow cytometry [26].

The therapeutic application of doxorubicin as a chemotherapeutic agent is associated with serious side effects on non-tumorigenic cells, with cardiovascular tissues especially sensitive [51]. The use of nanoparticles as carriers is expected to lead to the result that the combination of low-dose DOX with Ag NPs provides effective cytotoxicity against cancer cells and minimizes side effects. Therefore, silver nanoparticles were synthesized using *Coffea arabica* extract. Subsequently, the cytotoxic effects of GS-Ag NPs in combination with DOX in MCF-7 breast cancer cells were assessed, and their effects on the normal heart cell line H9c2 were investigated. For this purpose, an MTT assay was performed. In addition, Annexin-V/PI staining and mRNA expression of Bax, Bcl2, and p53 were performed to measure the apoptosis process. GS-AgNPs revealed lower cytotoxicity against normal cells and higher cytotoxicity against cancer cells compared with chemical nanoparticles. The combination of 20 μM Ag NPs/0.3 μM DOX revealed a cytotoxic effect against cancer cells with low toxic effects on normal cells. No significant changes were observed in cell migration capacity, apoptosis, and gene expression of BAX, Bcl-2, and p53 compared with the untreated control [52].

### 4.2. Antibacterial Activity

#### Waste Materials

Silver nanoparticles might find application in medicine since they possess the ability to inhibit both pathogenic and spore-forming bacteria. They can be obtained by various methods that are harmful to the environment. Alternatively, we can use green synthesis from agri-food waste. Green synthesis of silver nanoparticles was performed using an aqueous extract of *Carthamus tinctorius* waste. Homogeneous, spherical particles with a diameter of approximately 8.67 nm were obtained. FT-IR spectroscopy confirmed that different functional groups were responsible for particle reduction and stabilization. The antimicrobial activity of the nanoparticles was tested on *Staphylococcus aureus* (G+) and *Pseudomonas fluorescens* (G-). The nanoparticles inhibited the growth of bacteria from a concentration of 0.9 μg/mL [53].

### 4.3. Alternative Biomedical Applications

#### Plants

An innovative method for the electrochemical detection of serotonin via Ag NPs and rGO nanocomposite was proposed. Organic biosynthesis using *Salvia rosmarinus* leaf extract for its creation was applied. The sensitivity of the developed sensor was evaluated by differential pulse voltammetry, from which a very low detection limit of 78 pM was obtained—A result not previously reported in the literature. The findings were successfully applied to serotonin determination in artificial urine samples [35].

Polysaccharides of sea sources have the potential to enhance wound healing [54]. AGO agarose oligosaccharides extracted from *Rhodophyta* were utilized as a reducing agent and stabilizer for the biosynthesis of Ag NPs, then successfully combined with odorranain-A—an antimicrobial peptide. Thus, a new composite nanomaterial AGO-Ag NPs-OA was obtained. Using TEM and a Malvern particle size analyzer, the authors found that AGO-Ag NPs-OA were spherical or elliptical with an average size of about 100 nm. CD spectroscopy showed that AGO-Ag NPs stabilized the α-helical structure of OA. The nanocomposite exhibited stronger antimicrobial activity than AGO-Ag NPs and biocompatibility and a significant wound-healing accelerating effect [55].

### 4.4. Multidirectional Activity

#### Plants

Paclitaxel is an organic chemical compound from the terpene alkaloid group known for its anticancer activity. It was first isolated from the *Taxus brevifolia* tree [56]. This research aimed to develop a platform that would possess the properties of both silver nanoparticles and paclitaxel with reduced toxicity. Ag NPs were synthesized by reducing an aqueous AgNO_3_ salt solution with an aqueous extract of *T. brevifolia* leaves without using a catalyst or surfactant. Nanoparticle shape and size determination was accomplished by using transmission electron microscopy and dynamic light scattering techniques. To investigate the cytotoxicity of Ag NPs, silver nitrate, and paclitaxel, an MTT assay on human MCF-7 cells and a microscopic method with DAPI fluorescence staining were used. The MTT proved that Ag NPs biosynthesized with *Taxus brevifolia* leaf extract exhibit the strongest suppressive effect on cancer cells. The antibacterial activity of the three substances was investigated for gram-positive bacteria—for example, *Staphylococcus aureus*—and gram-negative bacteria—*Escherichia coli* and *Pseudomonas aeruginosa*. The IC_50_ was 3.1 mM for paclitaxel, compared with 1.5 mM for the new Ag NPs. Paclitaxel exhibited no antimicrobial activity, while Ag NPs revealed dose-dependent antimicrobial activity (MIC: 1.6 nM for gram-positive bacteria and 6.6 mM for gram-negative bacteria) compared with silver nitrate solution (MIC: 1.5 and 6.2 mM, respectively) [33].

Compounds in *Allium cepa* shells may be a reductant in the biosynthesis of Ag NPs. [57] A 1 mmol solution of AgNO_3_ was added dropwise to an aqueous extract of onion shells. The mixture was heated at 90 °C for 30 min. A color change in the mixture occurred during the process. Using this method, Ag NPs were synthesized with an average size of 12.5 nm, which was confirmed by transmission electron microscopy. Thus synthesized Ag NPs can be applied as catalysts to promote the Knoevenagel and Hantzsch reactions. Furthermore, the Ag NPs were found to be reusable. After five consecutive runs, the average efficiencies for both transformations were between 91% and 94%, indicating bio-Ag NPs might find application as catalysts in the synthesis of new drugs. Ag NPs also exhibited acceptable antioxidant activity [58].

### 4.5. Toxicity

#### Plants

The toxic effects of silver nanoparticles synthesized by *Moringa oleifera* extract on oxidative stress biomarkers of *Oreochromis niloticus* were investigated. Additionally, the role of different types of selenium in counteracting this toxicity was tested. Fish were exposed to Ag NPs at sub-lethal concentrations, selenium nanoparticles were assayed at equivalent levels, and the antagonistic effect was verified for 2 and 4 weeks. The effect was investigated by monitoring oxidative stress marker levels—such as DNA fragmentation, lipid peroxidation LPO, catalase CAT, and superoxide dismutase SOD. There was a statistically significant increase in LPO and DNA fragmentation and a statistically significant decrease in CAT and SOD in the groups exposed to silver nanoparticles compared with the control group. Biosynthesized Se NPs and selenium ions revealed a positive role in detoxification after exposure to Ag NPs. Silver nanoparticles exhibit toxicity reflected by oxidative stress markers. The addition of Se NPs provides effective neutralization of their toxic effects [59].

## 5. Palladium Nanoparticles

### 5.1. Anticancer Activity

#### Plants

Pd NPs were designed and synthesized via an environmentally safe method, and the efficacy of the nanoparticles was evaluated on the human ovarian cancer line A2780. To observe the process, UV-Vis spectroscopy was used. Fourier transform infrared spectroscopy confirmed the role of *Evolvulus alsinoides* leaf extract as a reducing and stabilizing agent in the synthesis of Pd NPs. DLS and TEM revealed that the nanoparticle size was approximately 5 nm. After 24-h exposure to Pd NPs, cell viability assays exhibited dose-dependent cytotoxicity of Pd NPs. These results were confirmed by lactate dehydrogenase assays, increased generation of reactive oxygen species, autophagy, altered mitochondrial membrane potential, and increased caspase 3 activation [60].

### 5.2. Antibacterial Activity

#### Plants

An attempt was made to synthesize biosynthesized Pd NPs with *Santalum album* leaf extract. Spherical morphology was confirmed by TEM analysis. The results of the XRD analysis proved that the Pd NPs possessed a cubic crystal structure. The FT-IR method was applied to identify the main functional groups of phytochemicals responsible for their ability to carry out the bioreduction of palladium salts. Investigation of the antimicrobial activity revealed that the Pd NPs obtained from *S. album* exhibited much stronger activity directed against Gram-negative than Gram-positive bacteria. The green synthesis method using *S. album* leaf extract for the synthesis of palladium nanoparticles is efficient and economical, offers the possibility of large-scale production transition, and the synthesized Pd NPs might be applied as antimicrobial agents [29].

The biosynthesis of palladium nanoparticles with a water extract of *Bauhinia variegata* bark was reported (Figure 3). FEG-SEM studies revealed an aggregated, irregular shape. The elemental composition and purity of the Pd NPs were analyzed using EDS. HR-TEM confirmed the size of Pd NPs between 2 and 9 nm. The effect of biosynthesized Pd NPs on the bacteria *Bacillus subtilis*, *Staphylococcus aureus*, and *Escherichia coli* was investigated. The palladium nanoparticles exhibited an excellent inhibition effect on the growth of *Bacillus subtilis*. Significant activity against the fungus *Candida albicans* was also observed. Antitumor studies were performed on the MCF-7 cell line. Compared with the reference drug, Pd NPs showed greater antitumor activity with an IC_50_ value of 41.37 µg/mL [61].

Considering the development of multidrug resistance in pathogenic bacteria and the prevalence of cancer, the application of Pd NPs derived from algae might provide a promising novel treatment option. Therefore, Pd NPs were synthesized by a one-step green method using an extract from the brown alga *Padina boryana* (PB extract), and their antimicrobial, inhibitory activity against bacterial biofilm formation and anticancer activity were evaluated. Pd NPs were characterized regarding size, shape, morphology, surface area, charge, crystal structure, and adsorption on the surface of Pd NPs of PB extract via different techniques. The results revealed that the palladium nanoparticles have an average size of 8.7 nm, have a crystalline structure, and exhibit a zeta potential of −28.7 ± 1.6 mV. FT-IR analysis revealed that the extract stabilizes Pd NPs with various functional groups derived from phenols, aliphatic hydrocarbons, and aliphatic amines. GC-MS analysis showed participation in the biosynthesis of 23 compounds out of 31 total compounds in the extract [62].

The effect of the synthesized nanoparticles was tested on bacteria—*Staphylococcus aureus*, *Aeromonas enteropelogenes*, *Acinetobacter pittii*, *Pseudomonas aeruginosa*, *Escherichia fergusonii*, and *Proteus mirabilis*. The authors observed that the biosynthesized Pd NPs exhibited antimicrobial activity against these bacteria and inhibited bacterial biofilm formation with a minimum inhibitory concentration between 62.5 and 125 μg/mL. Furthermore, cell viability assays were performed, which revealed concentration-dependent cytotoxicity toward MCF-7 breast cancer cells. Pd NPs stimulated the apoptosis process, which was found by increasing the mRNA expression of marker genes such as p53, whose stimulation increased 5.5-fold; the expression of Bax protein and caspase-3 increased three-fold, while the expression of caspase-9 increased two-fold at the tested concentration of 125 μg/mL [63].

### 5.3. Alternative Biomedical Applications

#### Plants

The biosynthesis of palladium Pd NP nanoparticles modified with reducing graphene oxide (rGO) sheets was conducted in a one-pot strategy using *Ficus carica* as the reducing agent. The nanoparticles were characterized by morphological and structural analyses—including ultraviolet spectroscopy, FT-IR, X-ray diffraction, and Raman spectroscopy. The results demonstrated that the nanoparticles are spherical, and their dimension is estimated to be approximately 0.16 nm. Pd NP/rGO exhibits high catalytic activity in Suzuki reactions under aqueous and aerobic conditions. The catalyst is reusable. The final product has the potential to make a significant contribution to the development of green chemistry in the synthesis of new drugs [64].

Nanoparticle biosynthesis was conducted with *Camellia sinensis* leaf extract as a reducing and stabilizing agent. The resulting catalyst Pd@B.tea NPs were characterized by spectroscopy. The Pd@B.tea NPs can be applied in Suzuki coupling reactions in an eco-friendly environment. It achieved significant production efficiency, and the catalyst was recycled seven times without remarkable loss of its catalytic activity [37].

*Hibiscus rosasinensis L.* extract was applied to synthesize palladium nanoparticles. MCaLig deposition was used to obtain PdNPs@Fe_3_O_4_- lignin. The magnetic NPs were used as catalysts in the Suzuki–Miyaura reaction. The catalytic performance was retained after seven cycles [38].

A facile and environmentally safe method was used to synthesize Pd NPs with an aqueous *Origanum vulgare* (OV) extract. Moreover, the compounds in the OV extract are not only responsible for the reduction but also act as a stabilizer, which was confirmed by spectroscopic methods. FT-IR investigation revealed that the extract also functionalizes the nanoparticles. Furthermore, the synthesized Pd NPs were successfully used as catalysts for the selective oxidation of alcohols, which might lead to the development of more eco-friendly approaches for the synthesis of new and currently existing therapeutic substances [39].

Pd NPs were recovered from industrial wastewater with a laser without reducing agents. Laser parameters—such as laser wavelength, power, and irradiation time—were optimized. Efficient recovery of palladium nanoparticles from industrial wastewater was achieved at a laser wavelength of 355 nm, power of 40 mJ/pulse, and irradiation time of 30 min. Pd NPs showed satisfactory catalytic activity in the reduction of 4-nitrophenol [36].

### 5.4. Toxicity

#### Plants

The toxicity of palladium nanoparticles synthesized chemically and using *Annona squamosa* seeds was evaluated. An animal model of zebrafish was used for this purpose. The nanoparticles created by green chemistry were characterized by UV-Visible, EDX, FTIR, zeta potential, and FE-SEM spectroscopy. *Danio rerio* was exposed to nanoparticles for 96 h at a concentration of 0.4 ng/L, and then oxidative stress markers (SOD, CAT, LPO) and the induced histological changes were examined (Figure 4). When exposed to chemically synthesized Pd NPs, all markers exhibited elevated levels. In the group exposed to As-Pd NPs, CAT activity gradually decreased up to 96 h compared with the control group. LPO, on the other hand, demonstrated an increase up to 72 h and a sudden decrease near 96 h. Histological lesions—such as ruptured hepatocytes, sinus congestion, and erythrocyte accumulation—were noted in both groups but significantly less intense in the As-Pd NPs group. As-Pd NPs synthesized by green chemistry are less toxic compared with chemically synthesized Pd NPs [65].

## 6. Platinum Nanoparticles

### 6.1. Anticancer Activity

#### Waste Materials

Platinum nanoparticles were biosynthesized with *Punica granatum* peel (Figure 3) and tested on the MCF-7 line. The prepared Pt NPs were extensively characterized. The efficacy of Pt NPs was determined by MTT viability assay, propidium iodide staining assay, and flow cytometry on the MCF-7 tumor cell line. The biosynthesized monodisperse platinum nanoparticles had an IC_50_ value of 17.84 µg/mL after 48 h of incubation. Propidium iodide staining revealed that the platinum nanoparticles induced apoptosis through DNA fragmentation [27].

### 6.2. Antibacterial Activity

#### Waste Materials

Single-step synthesis of platinum nanoparticles with an extract from the rind of *Garcinia mangostana* fruit was attempted. The formation of platinum nanoparticles was achieved after heating the solution for 20 min at approximately 80 °C. Parameters—including contact time, temperature, and pH—were optimized to obtain stable and homogeneous nanoparticles. The synthesized platinum nanoparticles were characterized. An attempt was made to evaluate the antibacterial activity of the synthesized nanoparticles before and after combination with commercially applied antibiotics in comparison with free antibiotics. Interestingly, Pt NPs combined with antibiotics exhibited enhanced antibacterial activity against pathogenic bacteria, suggesting the occurrence of synergism between nanoparticles and antibiotics [30].

### 6.3. Alternative Biomedical Applications

#### Plants

A platinum nanocomposite stabilized by *Momordica charantia* polysaccharide (BGP) was prepared. Nanoparticles were formed by a reaction of BGP with the platinum salt K_2_PtCl_4_. Pt-BGP NPs catalyzed the decomposition of H_2_O_2_, which could oxidize TMB, indicating that they possess peroxidase activity. The reaction kinetics followed the Michaelis-Menten formula. Colorimetric detection of ascorbic acid was attempted with Pt-BGP NPs, which revealed high selectivity and sensitivity with the detection of 0.191 μM. With Pt-BGP NPs, the measurement precision of the samples was nearly 99%. Pt-BGP NPs have great potential for application in the colorimetric detection of ascorbic acid [66].

Platinum salts were effectively reduced to their corresponding Pt NPs in the presence of an aqueous extract of *Phoenix dactylifera* type Zahidi—a prolific source of polyphenols [67]—leading to the reduction of Pt^4+^ to Pt^0^ atoms. TEM analysis revealed that the Pt NPs have a size in the range of 30–45 nm. The images obtained by the FE-SEM technique showed the nanoparticles have a spherical shape. XRD diffraction study revealed that Pt NPs have a crystalline structure. FT-IR spectrum showed peaks, which were used to identify the functional groups responsible for the reduction and stabilization of the nanoparticles. The effect of the synthesized nanoparticles was assayed on SKO-3 ovarian, and SK-GT-4 esophageal cancer cell lines and the rate of cell growth inhibition was measured for 72 h. The cytotoxicity assay demonstrated a strong toxic effect on cancer cells. The effect of nanoparticles on the gram-negative bacterial strain *Pseudomonas aeruginosa* and the gram-positive bacterial strain *Streptococcus pyogenes* was also investigated. The results revealed statistically significant inhibitory activity directly proportional to the concentration [68].

### 6.4. Toxicity

#### Plants

Efforts were made to investigate the effect of biosynthesized Pt NPs on human embryonic kidney HEK293 cells. Nanoparticles were synthesized using *Azadirachta indica* L. leaf extract and characterized using dynamic light scattering and transmission electron microscopy. MTS cytotoxicity assay and neutral red uptake assay were performed. Cell viability decreased in a manner in direct proportion to concentration and duration, and an increase in the generation of reactive oxygen species was noticed. Along with the decrease in cell survival, an increase in caspase 3 DNA fragmentation and significantly altered mitochondrial membrane potential were observed. Thus, it can be concluded that mitochondria are an important target for the generation of platinum nanoparticle toxicity in the HEK293 line. Moreover, an increase in oxidative stress markers—such as lipid peroxidation—was observed. Notably, higher susceptibility to nanoparticles was observed after 24 h, which corresponded to a stronger stimulation of apoptosis [69].

## 7. Copper Nanoparticles

### 7.1. Anticancer Activity

#### Plants

An optimal method for the biosynthesis of copper nanoparticles using *Allium noeanum* leaf extract was established. Cu NPs were assigned to be screened on human endometrial cancer lines (Ishikawa, HEC-1-A, HEC-1-B, and KLE). The antioxidant activities were examined by DPPH assay. IC_50_ values of *A. noeanum* aqueous extract and copper nanoparticles against the HEC-1-B cell line were 548 and 331 mg/mL, respectively; against HEC-1-A line, were 583 and 356 mg/mL, respectively; against the KLE line, were 609 and 411 mg/mL; and against Ishikawa cell line were 560 and 357 mg/mL [28]. IC_50_ values indicate the low efficacy of copper nanoparticles biosynthesized using *A. noeanum* extract.

### 7.2. Antibacterial Activity

#### Plants

The ability of *Ziziphus spina-christi* fruit extracts as reducing agents in the green synthesis of Cu NPs was investigated. Tests were performed to confirm that the Cu NPs could be used as an adsorption nanomaterial to remove crystal violet from an aqueous solution. The results revealed that 95% of the substance with an adsorption capacity of approximately 37.5 mg/g was removed using a small amount of adsorbent in a relatively short time. Furthermore, the antimicrobial activity of Cu NPs was investigated against two bacteria: *Escherichia coli* and *Staphylococcus aureus*. Cu NPs biosynthesized by *Z. spina-christi* fruit extract at high concentrations and methanol extract exhibit high antimicrobial activity against both microbial species [31].

### 7.3. Alternative Biomedical Applications

#### Plants

Cu NPs were biosynthesized with *Plantago asiatica* leaf extract under environmentally friendly reaction conditions. The reaction progress was monitored using UV-Vis spectroscopy. The catalytic activity of Cu NPs was evaluated by aldehyde cyanidation. This method has numerous benefits, including reduced reaction time and higher efficiency [70].

The synthesis of copper nanoparticles (Cu NPs) on manganese dioxide NPs was carried out using *Centella asiatica* leaf extract without stabilizers and surfactants. The phenolic hydroxyl groups in the extract are considered to reduce copper ions in the solution, which are then stabilized on the surface of manganese dioxide nanoparticles. The composite material was found to act as a catalyst capable of reducing Congo red, methylene blue, rhodamine B and 2,4-dinitrophenylhydrazine or 4-nitrophenol in an aqueous medium at room temperature. The high stability of the Cu/MnO_2_ nanocomposite allows the catalyst to be used repeatedly without loss of activity [71].

Extracts from the aboveground parts of *Euphorbia maculata* were used to prepare Ni@Fe_3_O_4_ and CuO NPs. The photocatalytic activity of the synthesized NPs was investigated in the degradation of different organic dye pollutants such as Congo red, methylene blue and Rhodamine B. A comparison of the photocatalytic activities of the biosynthesized nanoparticles reveals that the activity of CuO NPs is higher than that of Ni@Fe_3_O_4_ NPs. The photocatalyst performance did not change significantly after four cycles, indicating excellent photocatalytic stability [40].

## 8. Other Transition Metal Nanoparticles

### 8.1. Anticancer Activity

#### 8.1.1. Plants

Magnetic iron oxide nanoparticles were synthesized by a one-step green synthesis using *Sargassum muticum* extract. Nanoparticles exhibited antitumor potential against leukaemia (Jurkat), breast cancer (MCF-7), cervical cancer (HeLa), and liver cancer (HepG2) cell lines with IC_50_ of 6.4 ± 2.3 µg/mL, 18.75 ± 2.1 µg/mL, 12.5 ± 1.7 µg/mL, 23.83 ± 1.1 µg/mL, respectively, after 72h exposure. Further analysis confirmed that magnetic iron oxide nanoparticles have the ability to induce apoptosis through the activation of caspase-3 and -9 and the accumulation of nanoparticles in the sub-G1 phase [72].

#### 8.1.2. Bacteria

Magnetic iron oxide nanoparticles (MIO NPs) were biosynthesized using the isolated supernatant of the bacterial Bacillus cereus strain HMH1. Analysis revealed that the average particle size was approximately 29.3 nm. The toxicity of the nanoparticles was assessed using the MTT assay on MCF-7 and 3T3 cell lines. The IC_50_ for the MCF-7 line > 5 mg/mL and IC_50_ for the 3T3 line > 7.5 mg/mL, and the effect is concentration-dependent [73].

### 8.2. Antibacterial Activity

#### 8.2.1. Plants

Nanoparticles of a combination of three metals (Cu/Cr/Ni) were prepared using the extract of the plant *Echinops persicus* in a simple, biocompatible, and non-toxic procedure. The synthesized nanoparticles were characterized by various analytical methods. Then, their catalytic activity in the synthesis of biologically active compounds was evaluated. Cu/Cr/Ni NPs were used as a catalyst for the preparation of quinolines and spirooxindoles under solvent-free conditions with satisfactory efficiency. The antimicrobial properties of copper nanoparticles were investigated against *E. coli*, *S. aureus*, and *B. cereus*. Cu/Cr/Ni nanoparticles have remarkable catalytic and antimicrobial activities [74].

#### 8.2.2. Bacteria

The formation of biofilms by bacteria is one of the key survival strategies. Bacteria can survive chemical and physical stress on biotic and abiotic surfaces. This presents an urgent need to develop novel agents against biofilms. Promising results were obtained by synthesizing titanium dioxide nanoparticles using the green chemistry method using an extract from *Carum copticum*. They were active against both Gram+ and Gram- bacteria. Over 70 percent inhibition of test bacteria biofilm formation in the presence of titanium oxide nanoparticles was recorded—colonization of the glass surface was significantly reduced. Furthermore, the nanoparticles inhibited the secretion of exopolysaccharides (EPS) by *E. coli* ATCC25922 and *P. aeruginosa* PAO1 bacteria by 62.08 and 74.94%, respectively. Additionally, TiO2-NPs could eliminate the formed biofilms of E. coli ATCC 25922 and *P. aeruginosa* PAO1 by 60.09 and 64.14%, respectively. The results indicate the effectiveness of titanium oxide nanoparticles synthesized using a plant extract in inhibiting and eradicating bacterial biofilms [75].

### 8.3. Alternative Biomedical Applications

#### Plants

The *Aedes aegypti* mosquito is responsible for spreading the dengue virus causing significant mortality and financial losses in most tropical regions. The use of synthetic insecticides results in insect resistance, environmental pollution, and health problems. TiO_2_ nanoparticles were produced using an aqueous extract of *Pouteria campechiana* leaves. SEM analysis showed that the synthesized TiO_2_ NPs were spherical. The concentration of 900 μg/mL of TiO_2_ NPs had excellent lethal activity against various larval and pupal stages of *A. aegypti*. The results revealed that the aqueous extract of *P. campechiana* leaves could reduce TiO_2_ to TiO_2_ NPs and could be considered a mosquito control agent [76].

## 9. Conclusions

Nanoparticles have the potential to be a breakthrough in medicine. Metallic NPs induce oxidative stress caused by their small size and reactive surface area, leading to cellular responses. Their root mechanism of action is the induction of apoptosis by generating ROS and cell death caused by oxidative stress. The hope for the development of nanoparticles in medicine is nanoparticles biosynthesized from natural extracts. They provide lower toxicity to regular cells and higher anti-tumor activity to cancerous cells. They also eliminate the toxicity associated with poisonous agents that stabilize the chemically synthesized nanoparticles. Gold and silver nanoparticles themselves have cytostatic and bactericidal properties, and the strength of their action depends on the natural extract used—i.e., substances adsorbed on the surface. Biosynthesized palladium, copper, and platinum nanoparticles might provide renewable catalysts useful in the synthesis of new drugs. Recently, there has been a focus on generating NPs to overcome biological barriers specific to diseases or groups of patients, leading to precision in medicine. Information such as patients’ environmental exposure, genetic profile, or co-existing diseases should be used to individualize the treatment [15]. Metallic nanoparticles synthesized with the use of bacteria, fungi, or plants have the potential to have multidirectional applications—from the environmentally safer synthesis of new substances to the application of NPs themselves in the fight against cancer or bacterial infections.

## Figures and Tables

**Figure 1 pharmaceutics-14-02286-f001:**
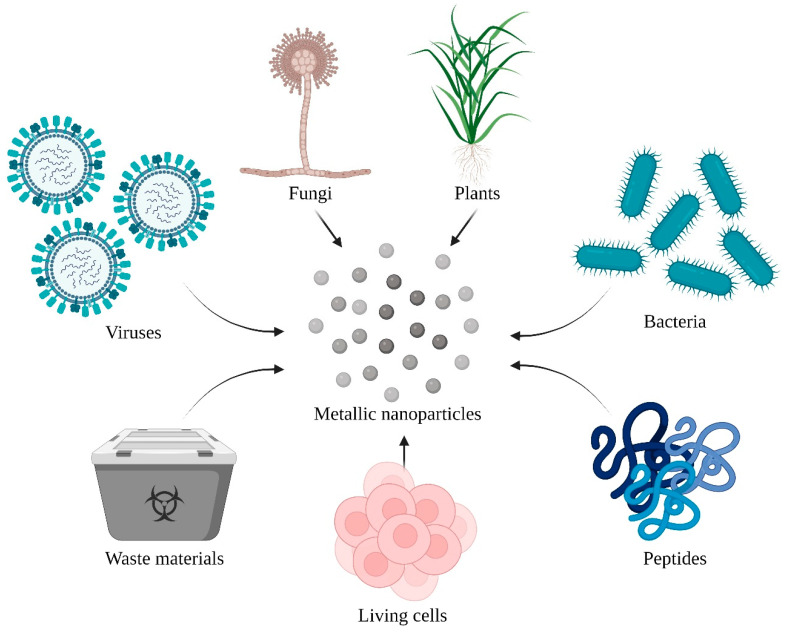
Various ways of achieving green synthesized nanoparticles. (Created with BioRender.com, accessed on 29 July 2022).

**Figure 2 pharmaceutics-14-02286-f002:**
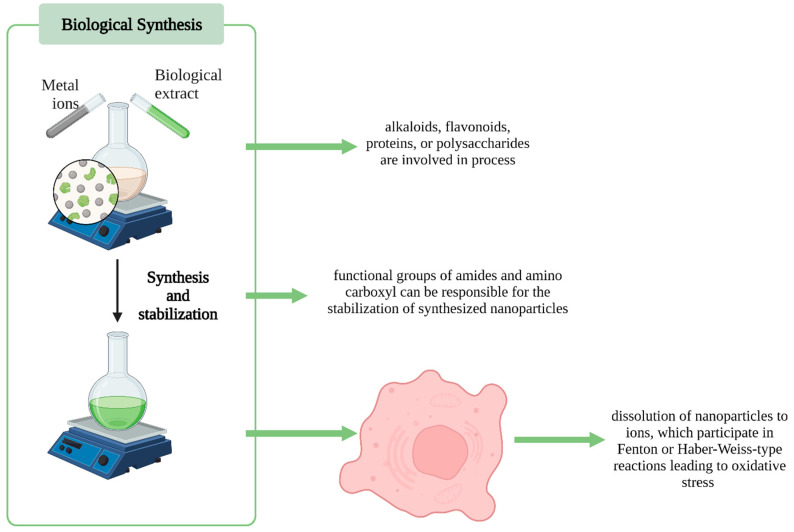
Summary of green nanoparticle synthesis (Created with BioRender.com, accessed on 29 July 2022).

**Figure 3 pharmaceutics-14-02286-f003:**
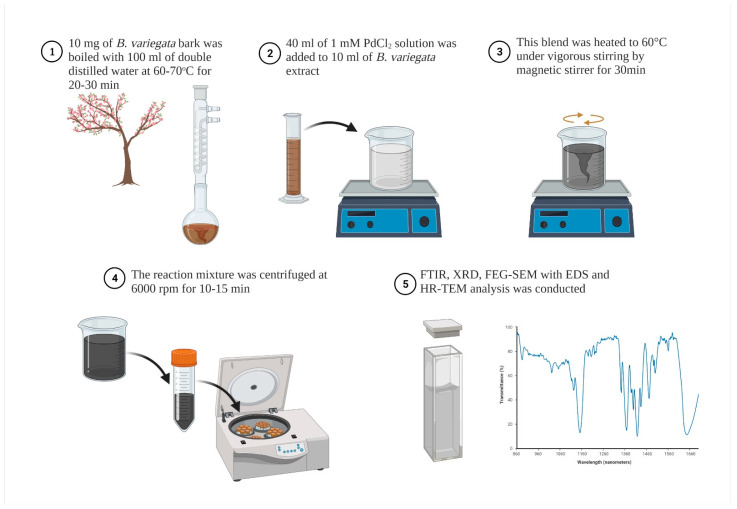
Biosynthesis of Pd NPs via *B. variegata* extract (Created with BioRender.com accessed on 29 July 2022).

**Figure 4 pharmaceutics-14-02286-f004:**
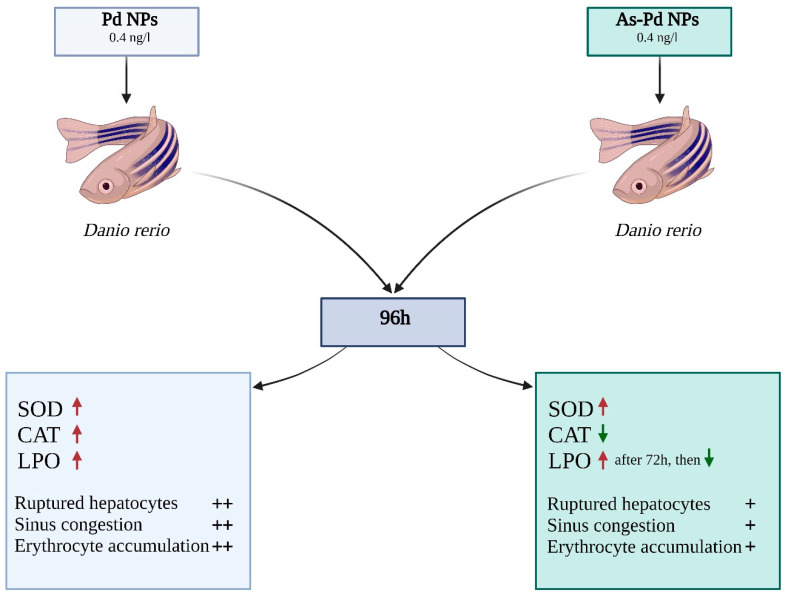
Comparison of the toxicity of palladium nanoparticles (Created with BioRender.com accessed on 29 July 2022).

**Table 1 pharmaceutics-14-02286-t001:** Summary of analysis of selected biosynthesized nanoparticles.

Biomedical Applications	Metal	Origin of the Extract	Analysis	Reference
Anticancer activity	Au	*Allium fistulosum*	FT-IR, zeta potential, DLS, XRD, UV-vis, SEM, and EDX	[22]
		*Cordyceps militaris*	FT-IR, HR-TEM, and XRD	[23]
		*Dendrobium officinale*	TEM, DLS, FT-IR, and EDX	[24]
		*Taxus baccata*	UV-Vis, TEM, and FT-IR	[25]
	Ag	*Matricaria recutita*	UV-Vis, TEM, XRD, EDX and FT-IR	[26]
	Pt	*Punica granatum*	UV-Vis, TEM, XRD, SEM and FT-IR	[27]
	Cu	*Allium noeanum*	EDX, FT-IR, FE-SEM, UV-Vis, TEM, and XRD	[28]
Antibacterial	Pd	*Santalum album*	UV-vis, XRD, TEM, and FTIR	[29]
	Pt	*Garcinia mangostana*	FT-IR, UV-Vis, XRD, zeta potential measurements, HR-TEM, and HR-SEM	[30]
	Cu	*Ziziphus spina-christi*	UV-Vis, FT-IR, FESEM, TEM and XRD	[31]
	Ag	*Bacillus sp.*	UV-Vis, TEM, XRD, EDS	[32]
Anticancer and antimicrobial activity	Ag	*Taxus brevifolia*	UV-Vis, XRD, SEM, DLS, TEM and FT-IR	[33]
	Au	Chitosan	UV-Vis, XRD, DLS, TEM and FT-IR	[34]
Alternative biomedical applications	Ag	*Salvia rosmarinus*	UV-Vis, FTIR, TEM, and EIS	[35]
Indirect biomedical applications	Pd	Industrial wastewater	XRD, FE-SEM, TEM, ICP-OES	[36]
		*Camellia sinensis*	UV-vis, XRD, FT-IR, FE-SEM, TEM, and EDS	[37]
		*Hibiscus Rosasinensis*	TEM, FE-SEM, VSM, FT-IR, XRD, EDS, and UV-Vis	[38]
		*Origanum vulgare*	UV-Vis, FT-IR, XRD, TEM, and TGA	[39]
	Ni, Cu	*Euphorbia maculata*	FT-IR, FE-SEM, XRD, UV-Vis, EDS, BET, TGA	[40]

## Data Availability

Not applicable.

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
