# Peer review of "Biosynthesized Gold, Silver, Palladium, Platinum, Copper, and Other Transition Metal Nanoparticles"

_pharmaceutics, 2022, doi:10.3390/pharmaceutics14112286_

Round 1

Reviewer 1 Report

The manuscript entitled “Biomedical Applications of Biosynthesized Gold, Silver, Palladium, Platinum, Copper, Iron Nanoparticles” is an attempt to reveal the application of metallic based nanoparticles. I have suggestion to authors for improvement of this review article. Below the issues which authors should address?

The title of the manuscript is not appropriate with the manuscript and the review is not structured as per its title. There are many issues in the review

1.       Since the authors mentioned the role of different Biosynthesized metallic nanoparticles so, there should be separate sections for each metal such as gold, silver etc and other nanoparticles should be discussed  in separate section.

2.       there should be mechanism of synthesis of each metallic particles and importance of biosynthesized nanoparticles since I can understand authors have mentioned in text about plants, bacteria, fungi based nanoparticles.

3.       There should be mechanism presented for the each nanoparticles how they are helpful in biomedical applications since lot of literature is available.

4.        For examples figure 1 is just a simple illustration synthesis of metal based nanoparticles which everyone knows in the field. There should be figure showing importance of green synthesis of nanoparticles against chemical synthesis.

5.       Section 2 describes Anticancer Activity of nanoparticles and section 3 describes its antibacterial activity. So why authors have another section 4 describing their role as Anticancer and Antimicrobial Activity of Biosynthesized Nanoparticles. This is totally confused.

6.       Table 1 has no references

7.       The authors should focus on application of nanoparticles not on the synthesis and they should show the mechanism with different references how metallic nanoparticles in bacteria are destroying the bacterial cell wall. Same should be in case with antifungal and anti-cancerous activity.

8.       The manuscript is not written in the flow this is very confusing.

9.       The figure 2, 3 and 4 are demonstrating the synthesis of nanoparticles there is no need for this instead of these figure they should replace with mechanism of action of these nanoparticles.

10.   Further many sentences are not clear.      

Overall the manuscript is not well written and is not suitable for the publication in present form.

Author Response

Response to the Reviewer 1:

According to the Reviewer’s suggestions:

  1. The title of the manuscript is not appropriate with the manuscript and the review is not structured as per its title. There are many issues in the review

We have changed the title to make it more appropriate.

  1. Since the authors mentioned the role of different Biosynthesized metallic nanoparticles so, there should be separate sections for each metal such as gold, silver etc and other nanoparticles should be discussed in a separate section.

As suggested, we have organized the chapters according to the metal type. In addition, we have kept the sections according to biomedical use and source of extract.

  1. there should be mechanism of synthesis of each metallic particles and importance of biosynthesized nanoparticles since I can understand authors have mentioned in text about plants, bacteria, fungi based nanoparticles.

We have included information in the introduction on biomedical applications of transition metal nanoparticles

  1. There should be mechanism presented for the each nanoparticles how they are helpful in biomedical applications since lot of literature is available.

We expanded the importance of transition metal nanoparticles in biomedical applications at https://doi.org/10.3390/ijms23126688.

  1. For examples figure 1 is just a simple illustration synthesis of metal based nanoparticles which everyone knows in the field. There should be figure showing importance of green synthesis of nanoparticles against chemical synthesis.

and

  1. The figure 2, 3 and 4 are demonstrating the synthesis of nanoparticles there is no need for this instead of these figure they should replace with mechanism of action of these nanoparticles.

We have removed the two illustrations of the biosyntesis steps and replaced them with an illustration summarising the biosynthesis mechanism in chapter 1 (Figure 2)

  1. Section 2 describes Anticancer Activity of nanoparticles and section 3 describes its antibacterial activity. So why authors have another section 4 describing their role as Anticancer and Antimicrobial Activity of Biosynthesized Nanoparticles. This is totally confused.

We have removed this section, in accordance with the recommendations. The subsections are now called 'Multidirectional Activity' to avoid confusing the reader

  1. Table 1 has no references

Thank you for drawing attention to this deficiency, a correction has of course been made.

  1. The authors should focus on application of nanoparticles not on the synthesis and they should show the mechanism with different references how metallic nanoparticles in bacteria are destroying the bacterial cell wall. Same should be in case with antifungal and anti-cancerous activity.

The main mechanism of action of transition metal nanoparticles, according to the papers we found, is the accumulation of NPs in mitochondria and lysosomes, where the acidic environment leads to the dissolution of nanoparticles into ions, which in Fenton or Haber-Weiss-type reactions generate ROS, responsible for the cellular response of the nanoparticles. Of course, we added a paragraph to the introduction on this matter.

  1. The manuscript is not written in the flow this is very confusing.

We hope, the manuscript is a better reading after the reorganization.

We would like to thank the Reviewer for the valuable comments and suggestions. Accordingly, we have revised and tried our best to improve the manuscript. We sincerely hope that the revised manuscript will meet your approval.

Reviewer 2 Report

The article entitled Biomedical Applications of Biosynthesized Gold, Silver,  Palladium, Platinum, Copper, Iron Nanoparticles" presents the biosynthesized NPs for different biomedical applications.
Although the article was written well, i would suggest rearranging the section headings as the arrangement lacks focus in terms of the type of NPs. First, biosynthesis procedures must be arranged as a separate section (Section 2). All the biological applications as a separate section.
Characterizations must be discussed well and detailed.
Also, suggest explaining the effect of morphology on the bioefficacy of NPs.

Author Response

Response to the Reviewer 2:

The authors would like to thank the Reviewer for taking the time to review our paper. According to the Reviewer’s suggestions:

  1. Although the article was written well, I would suggest rearranging the section headings as the arrangement lacks focus in terms of the type of NPs. First, biosynthesis procedures must be arranged as a separate section (Section 2). All the biological applications as a separate section. Characterizations must be discussed well and detailed.

Following the recommendation, we have rewritten the entire paper, in which we have separated sections related to the type of metal, where we have made subsections for specific types of biomedical applications. In addition, we separated a section with the general mechanism of synthesis and detailing the relevance of NPs size studies (Chapter 1). We have also added a figure summarizing the synthesis and likely mechanism of action of NPs. If the reviewer feels that these topics should be developed and deepened, we are of course open to any suggestions.

Also, suggests explaining the effect of morphology on the bioefficacy of NPs.

We added information about the effect of morphology on the biological properties of nanoparticles in Chapter 1. For chemically synthesized transition metal nanoparticles, we have included information on the influence of the morphology of NPs in the paper https://doi.org/10.3390/ijms23126688.

The authors would like to thank the Reviewer for the thorough analysis of the paper and the valuable comments that significantly increased the scientific value of the article. We hope that the revised paper will meet your approval.

Round 2

Reviewer 1 Report

Authors have improved the manuscript so have a minor correction related to title. the title should be Biosynthesized Gold, Silver, Palladium, Platinum, Copper,  And Other Transition Metal Nanoparticles and their role in biomedical applications.